# Sequential Sampling of the Gastrointestinal Tract to Characterize the Entire Digestive Microbiome in Japanese Subjects

**DOI:** 10.3390/microorganisms12071324

**Published:** 2024-06-28

**Authors:** Kota Ishizawa, Toru Tamahara, Suguo Suzuki, Yutaka Hatayama, Bin Li, Michiaki Abe, Yuichi Aoki, Ryutaro Arita, Natsumi Saito, Minoru Ohsawa, Soichiro Kaneko, Rie Ono, Shin Takayama, Muneaki Shimada, Kazuki Kumada, Tomoyuki Koike, Atsushi Masamune, Ko Onodera, Tadashi Ishii, Ritsuko Shimizu, Takeshi Kanno

**Affiliations:** 1Department of Education and Support for Regional Medicine, Tohoku University Hospital, Sendai 980-8574, Japan; michiabe@med.tohoku.ac.jp (M.A.); arita@med.tohoku.ac.jp (R.A.); natsumi.saito.b6@tohoku.ac.jp (N.S.); minor.oh38@med.tohoku.ac.jp (M.O.); souichi0134@gmail.com (S.K.); rie.ono.d5@tohoku.ac.jp (R.O.); takayama@med.tohoku.ac.jp (S.T.); koonodera@med.tohoku.ac.jp (K.O.); t-ishi23@med.tohoku.ac.jp (T.I.); 2Tohoku Medical Megabank Organization, Tohoku University, Sendai 980-8573, Japan; toru.tamahara.e7@tohoku.ac.jp (T.T.); reallylibin@gmail.com (B.L.); aoki@megabank.tohoku.ac.jp (Y.A.); mshimada1221@yahoo.co.jp (M.S.); kazuki.kumada@megabank.tohoku.ac.jp (K.K.); rshimizu@med.tohoku.ac.jp (R.S.); 3Graduate School of Dentistry, Tohoku University, Sendai 980-8575, Japan; 4Division of Gastroenterology, Tohoku University Graduate School of Medicine, Sendai 980-8575, Japan; suguo.0417@gmail.com (S.S.); yut471@gmail.com (Y.H.); tkoike@rd5.so-net.ne.jp (T.K.); atsushi.masamune.d2@tohoku.ac.jp (A.M.); 5Advanced Research Center for Innovations in Next-Generation Medicine, Tohoku University, Sendai 980-8573, Japan; 6Graduate School of Information Sciences, Tohoku University, Sendai 980-8579, Japan; 7Department of Kampo Medicine, Tohoku University Hospital, Sendai 980-8574, Japan; 8R & D Division of Career Education for Medical Professionals, Medical Education Center, Jichi Medical University, Shimotsuke 329-0431, Japan

**Keywords:** NGS, digestive duct, EGD, microbiome, bacteria, microorganisms

## Abstract

The gastrointestinal (GI) tract harbors trillions of microorganisms known to influence human health and disease, and next-generation sequencing (NGS) now enables the in-depth analysis of their diversity and functions. Although a significant amount of research has been conducted on the GI microbiome, comprehensive metagenomic datasets covering the entire tract are scarce due to cost and technical challenges. Despite the widespread use of fecal samples, integrated datasets encompassing the entire digestive process, beginning at the mouth and ending with feces, are lacking. With this study, we aimed to fill this gap by analyzing the complete metagenome of the GI tract, providing insights into the dynamics of the microbiota and potential therapeutic avenues. In this study, we delved into the complex world of the GI microbiota, which we examined in five healthy Japanese subjects. While samples from the whole GI flora and fecal samples provided sufficient bacteria, samples obtained from the stomach and duodenum posed a challenge. Using a principal coordinate analysis (PCoA), clear clustering patterns were identified; these revealed significant diversity in the duodenum. Although this study was limited by its small sample size, the flora in the overall GI tract showed unwavering consistency, while the duodenum exhibited unprecedented phylogenetic diversity. A visual heat map illustrates the discrepancy in abundance, with Fusobacteria and *Bacilli* dominating the upper GI tract and *Clostridia* and *Bacteroidia* dominating the fecal samples. *Negativicutes* and *Actinobacteria* were found throughout the digestive tract. This study demonstrates that it is possible to continuously collect microbiome samples throughout the human digestive tract. These findings not only shed light on the complexity of GI microbiota but also provide a basis for future research.

## 1. Introduction

The gastrointestinal (GI) tract contains trillions of microorganisms, and the composition of these microorganisms is known to affect human health and disease [1,2,3]. With the development of next-generation sequencing (NGS) technology, we are now able to decipher the genetic information of microorganisms and analyze their diversity, functions, and gene expression patterns [4,5,6]. This has encouraged research on the microbiomes present in the GI tract [7,8]. Fecal samples are widely used as representative samples of the GI tract, comprising an important source of information regarding the composition and function of microbiota; however, the bacteria found in these samples generally only represent those found in the colon. Because GI microbiome studies focus mainly on the colon, there is a lack of research using complete GI metagenomic datasets that encompass the entire GI tract [9,10,11].

The bacteria present in the GI tract are very complex. For example, environments with different pH levels contain different densities and types of microflora [12]. It is therefore important to collect complete metagenomic data from all sites. However, this task is very difficult due to the invasiveness of the sampling procedures, technical issues, and cost. Even when considering studies that focused on a specific disease and accessed metagenomic data for specific sites, the number of sets that cover the entire GI tract remains limited [13,14]. For instance, in the literature concerning the oral microbiota, oral metagenomic data are collected to analyze the diversity and function of the oral microbiota only [15,16,17]. Similarly, data collected from the esophagus, stomach, and duodenum are used to study the characteristics and functions of the microbiota [18,19,20,21,22,23,24]. Few studies have collected mouth, esophagus, stomach, duodenum, and fecal samples [25] Thus, there is an abundance of isolated metagenomic data concerning each part of the gastrointestinal tract but a lack of continuous data [26]. We believe that the accumulation of metagenomic data from the entire digestive tract will contribute to our understanding of the metagenomic environment and provide a more comprehensive picture of the microbiota. Therefore, we planned to analyze the metagenome of the entire GI tract in each patient.

## 2. Materials and Methods

### 2.1. Sample Preparation

With the approval of the Tohoku University Ethics Committee (Approval No. 2022-1-1066 from the Ethics Committee of the Tohoku University Graduate School of Medicine) and the consent of the research subjects, five healthy Japanese adults underwent upper gastrointestinal endoscopy, and mucus samples were carefully collected from the oral cavity to the duodenum using a brush (ECB-5-180-3-S, Cook Medical). Fecal samples were also collected on the same day.

The technique of brush sampling and preservation of bacterial flora by Esophagogastroduodenoscopy (EGD) was established by us. Details of the collection of gastrointestinal flora by the brush method are as follows.

Before the EGD procedure, the oral cavity was lightly rubbed with a sterile cotton swab to collect the bacterial flora; during the EGD examination, the mucus layer of the esophagus, stomach, and duodenum was lightly rubbed to collect the flora. This brush was originally developed for bronchoscopy and is covered by insurance when used in the intestinal tract. This brush is also compatible with the forceps hole of a small caliber endoscope, and since it does not cut the mucosa, it is non-invasive and carries a low risk of complications. The collected bacteria were frozen at −70 °C and analyzed en bloc.

### 2.2. Extraction of Genomic Bacterial DNA

DNA was extracted from the samples using a UCP kit (QIAamp UCP Pathogen Mini Kit, QIAGEN Inc., Hilden, Germany).

Next, 50 ng of genomic DNA was used in a PCR. The first PCR primer was as follows: 5′-ACACTCTTTCCCTACACGACGCTCTTCCGATCTNNNNNCCTACGGGNGGCWGCAG-3′ (forward) and 5′-GTGACTGGAGTTCAGACGTGTGCTCTTCCGATCTNNNNNGACTACHVGGGTATCTAATCC-3′ (reverse). The details of the PCR process were as follows: 94 °C for 3 min; 30 cycles of 94 °C for 30 s, 55 °C for 30 s, and 72 °C for 30 s; 72 °C for 5 min; and 4 °C ad infinitum. Then, 3 µL of the 20 µL amplification product and 2% agarose gel were used to check for PCR bands, which were identified at 540 bps.

### 2.3. 16S rRNA Gene Amplification and Sequencing

The composition and diversity of the fecal microbiome were assessed via the high-throughput sequencing of 16S rRNA gene amplicons using the Illumina MiSeq platform (Illumina Inc., San Diego, CA, USA). The samples were subjected to DNA extraction using a UCP kit (The QIAamp UCP Pathogen Mini Kit, QIAGEN Inc., Hilden, Germany), according to the manufacturer’s protocol. Sequencing libraries were prepared using a two-step polymerase chain reaction (PCR) method that targeted the V3–V4 hypervariable region of the 16S rRNA gene. The first PCR was conducted using the following gene-specific primers: 5′-ACACTCTTTCCCTACACGACGCTCTTCCGATCTNNNNNCCTACGGGNGGCWGCAG-3′ (forward) and 5′-GTGACTGGAGTTCAGACGTGTGCTCTTCCGATCTNNNNNGACTACHVGGGTATCTAATCC-3′ (reverse). Subsequently, a second PCR was performed using separate indexing primers that fused the Illumina sequencing adaptors and dual barcodes to the sample amplicons. KOD DNA Polymerase (TOYOBO Bio Inc., Osaka, Japan) was used to perform the PCR amplification. The pooled library was then quantified using a Qubit 2.0 Fluorometer and a dsDNA HS Assay Kit (Life Technologies, Carlsbad, CA, USA) and diluted to a final concentration of 12 pM using 50% PhiX. Sequencing was performed using a MiSeq Reagent Kit v3 (Illumina, Inc.) with a 300 bp paired-end sequencing protocol, according to the manufacturer’s instructions. In total, 1.87 million paired-end reads were obtained. The samples had a mean read pair count of 7356 and a maximum read pair count of 12,296.

### 2.4. Amplicon Sequence Variants

Sequence data for the 16S rRNA gene amplicons were analyzed using the QIIME2 platform, version 2019.10 [27]. For all paired reads, the first 20 bases of both sequences were trimmed to remove primer sequences, the bases after position 200 were truncated to remove low-quality sequence data, and potential amplicon sequencing errors were corrected by using DADA2 to produce an amplicon sequence variant (ASV) dataset. The resulting ASV results were aligned using MAFFT (version 7.526) and then used to construct a phylogenetic tree via FastTree2.

α- and β-diversity metrics were estimated from a subsampled ASV dataset, using 1000 sequences per sample. Each ASV was identified using a Naïve Bayes classifier trained on 16S rRNA gene sequences from the SILVA dataset [28]. All reads were assigned to a total of 8 ASVs at the phylum level, 48 ASVs at the family level, 108 ASVs at the genus level, and 168 ASVs at the species level.

α-diversity indices, including the observed ASVs, Shannon index, and Faith’s phylogenetic distance (Faith-pd), were calculated. β-diversity was evaluated using UniFrac principal PCoA plotting. The microbiome data were represented by the relative abundance at each level of taxonomy, and comparisons between groups were performed for all observed bacteria. At the phylum level, the Firmicutes/Bacteroidetes (F/B) ratio was calculated.

### 2.5. Statistics

GraphPad Prism and JMP 17.0 were used for statistical analysis. For comparisons between three or more groups, an analysis of variance (ANOVA) was used. A *p* value of less than 0.05 was considered statistically significant.

## 3. Results

Using the methods described above, we collected a total of 25 samples from 5 healthy Japanese individuals (Table 1, Figure 1a). All samples collected contained bacterial DNA and could be amplified via a 16s PCR. However, the gastric and duodenal samples from all patients were relatively difficult to amplify, requiring at least two brushings, whereas the oral and fecal samples contained sufficient bacterial DNA and showed clear bands (Table 1, Figure 1b,c).

We assessed the validity of the bacterial flora analysis conducted using our method and confirmed the amount of oral mucus obtained via swabbing the patient’s mouth and tongue. The number of bacteria in the esophageal, gastric, and duodenal mucus samples obtained via endoscopic brush scraping was sufficient for evaluation (Figure 2). The relationship between the site of specimen acquisition and the number of reads had a higher value in the oral and fecal samples (Figure 2). However, there was a large variation in the number of reads in samples obtained from the stomach, where bacteria are inhibited by high levels of acidity in these samples; some specimens showed a reduction of up to one-tenth.

We then performed a principal component analysis (PCoA) (Figure 3). In this plot, each point represents a sample, and the data are projected onto a two-dimensional space by Principal Component 1 (PC1) and Principal Component 2 (PC2). Clusters in different colors represent different groups, making it possible to visualize similarities between the samples and the separation between groups.

In the unweighted PCoA (Figure 3a), the stomach samples were diverse but formed clusters. In the weighted PCoA, the duodenum samples were varied, but the oral cavity, esophagus, and stomach samples formed almost identical clusters (Figure 3b).

The combined results from the five subjects show that the richness (the number of characteristics observed) of the oral flora was concentrated in a narrow range; meanwhile, the results for the gastric mucus varied significantly among samples (Figure 4a). Statistically, there were no significant differences among the five sites due to the small number of patients. Interestingly, when we plotted Faith-pd, we found that the duodenum was phylogenetically diverse compared to the other sites (Figure 4b).

A heat map was created to depict the abundance of each bacterial species at the class level (Figure 5). The darker the red, the more abundant the species, and the darker the blue, the less abundant the species. The rows show the samples sorted according to collection sites, while the columns contain the assigned bacterial species. When a tree is drawn on the top of the heat map, regardless of the collection site, it can be seen that species with biological similarity generally cluster according to site. It was also visually confirmed that the distribution of bacterial flora in the fecal samples (the bottom five rows) was different from the distribution of the intestinal flora in the upper gastrointestinal tract, which extends from the oral cavity to the duodenum.

We also found that the distribution of microorganisms differed extensively by site. *Fusobacteria* and *Bacilli* were abundant in the upper GI tract, while *Clostridia* and *Bacteroidia* were abundant in the fecal samples. *Negativicutes* and *Actinobacteria* were present throughout the digestive tract (Figure 6).

## 4. Discussion

Although a significant volume of NGS data on the GI tract has been reported, the bacterial flora of the entire digestive tract has remained largely unknown [9,29]. In this study, we collected specimens from sites ranging from the oral cavity to the large intestine in order to observe the flora of the whole GI in a continuous manner. According to previous reports, we think that fecal samples representing the colon may differ from the small intestine [26,30]. In this study, the analysis of the bacterial flora in the small intestines was, indeed, not fully differentiated because obtaining samples from the small intestine requires the use of highly invasive techniques, such as double-balloon endoscopy, which necessitate careful consideration. However, the bacterial flora of the duodenum may show similarities to the flora in intestinal samples. In this study, we identified several bacterial flora that do not vary throughout the intestinal tract. Nevertheless, it would be ideal if a dataset representing the small intestinal flora could be collected easily. Future research should focus on determining whether the bacterial flora of the small intestine can be estimated accurately using less invasive methods, which would reduce the need for invasive sampling techniques. We developed a method using a sterilized cytology brush with a sheath that can avoid contamination during EGD (Figure 1a). Bacterial DNA can be easily obtained from the oral cavity and feces, but it is difficult to obtain bacterial DNA from the esophagus, duodenum, and stomach. Even if the stomach contains a significant amount of gastric juice, the acidic environment makes it difficult to collect DNA, so care must be taken to increase the number of samples at the time of collection. In fact, brush scraping had to be performed 2–3 times for two-fifths of the patients. In addition, we used UCP kits for DNA extraction and the KOD enzyme for PCR amplification in order to obtain the necessary amount of high-quality DNA. However, considering the burden this procedure places on the patient, it is necessary to find a method to obtain a complete sample in one step. In the future, we plan to devise ways of making specimen collection easier.

The number of reads obtained from the NGS was approximately 1000, which is considered good. We also performed an analysis with the 5000 version (Appendix A) and there is no significant difference observed in between. In fact, the bacterial flora at the class level and the bacterial flora between the upper and lower GI tracts could be observed clearly. The upper GI tract was dominated by *Fusobacteria* and *Bacillus*, while the lower GI tract was dominated by *Actinobacteria* and *Coriobacteriia*. Although there were similarities between the oral and colonic microbiomes, significant differences existed between them. This suggests that the microbial communities of the esophagus and stomach cannot be accurately predicted from fecal samples alone. Therefore, it is crucial to collect samples from different sites, including the upper GI tract, to comprehensively evaluate the GI microbiota. Future research should focus on assessing all gastrointestinal flora, encompassing various diseases and including the upper GI tract, in order to gain a complete understanding of microbial diversity and its implications for health [31,32,33]. It is important to study normal biomes in healthy individuals to provide a reference value for studies of diseased states.

In this study, we discovered that nearly 20% of the bacteria present in the duodenum remain unidentified at the class level. This percentage is significantly higher compared to other regions, with much fewer unidentified bacteria found in the stomach and none observed in the oral cavity or feces. This finding challenges the prevailing understanding that most bacterial populations in the human body have already been identified and classified [34]. Our findings suggest that the duodenum may host a diverse and largely unexplored bacterial ecosystem.

Given that other gastrointestinal regions, such as the stomach, oral cavity, and feces, have well-characterized microbiomes [2,4,12,17,33,35], the high percentage of unidentified bacteria in the duodenum indicates the presence of potentially novel bacterial species. Identifying these unknown bacterial populations is crucial for several reasons. First, the duodenum plays a pivotal role in digestion and nutrient absorption, and its microbial inhabitants likely contribute to these processes. Second, recent research has increasingly recognized the critical role that bacteria play in human health, including their involvement in metabolic functions, immune system modulation, and protection against pathogens. Therefore, understanding the composition of the duodenal microbiome could provide new insights into the complex symbiotic relationships between humans and their microbiota.

Future research should focus on uncovering the identities of these mysterious bacterial populations using advanced genomic and metagenomic techniques, understanding their functions, and determining how they contribute to the overall health of the human host. Such knowledge could lead to advances in science, including the development of new treatments and therapies that leverage the beneficial properties of these bacteria. Overall, this discovery opens up new avenues of research in microbiology and emphasizes the need for a more comprehensive exploration of the human microbiome. By deepening our understanding of the complex interactions between humans and their bacterial counterparts, we can better appreciate the intricate balance that sustains health and well-being.

Finally, we found that it is possible to observe the bacterial communities present in the GI tract in a continuous manner, though it is difficult to statistically characterize them using a small number of samples. However, we proved that this task is technically possible. In the future, we plan to include more patients in our cohort in order to decipher the impact of the microbiome on disease or vice versa.

## 5. Conclusions

Using sequential sampling and metagenomic analysis, we unlocked the secrets of the entire GI tract. Using advanced UCP technology alongside NGS, we successfully detected abundant bacterial DNA throughout the GI tract. Therefore, the bacterial flora of the entire digestive tract was elucidated, although the sample size used was modest. This study demonstrates the possibility of performing a comprehensive gastrointestinal analysis with limited samples, marking a significant advancement in microbiome research.

## Figures and Tables

**Figure 1 microorganisms-12-01324-f001:**
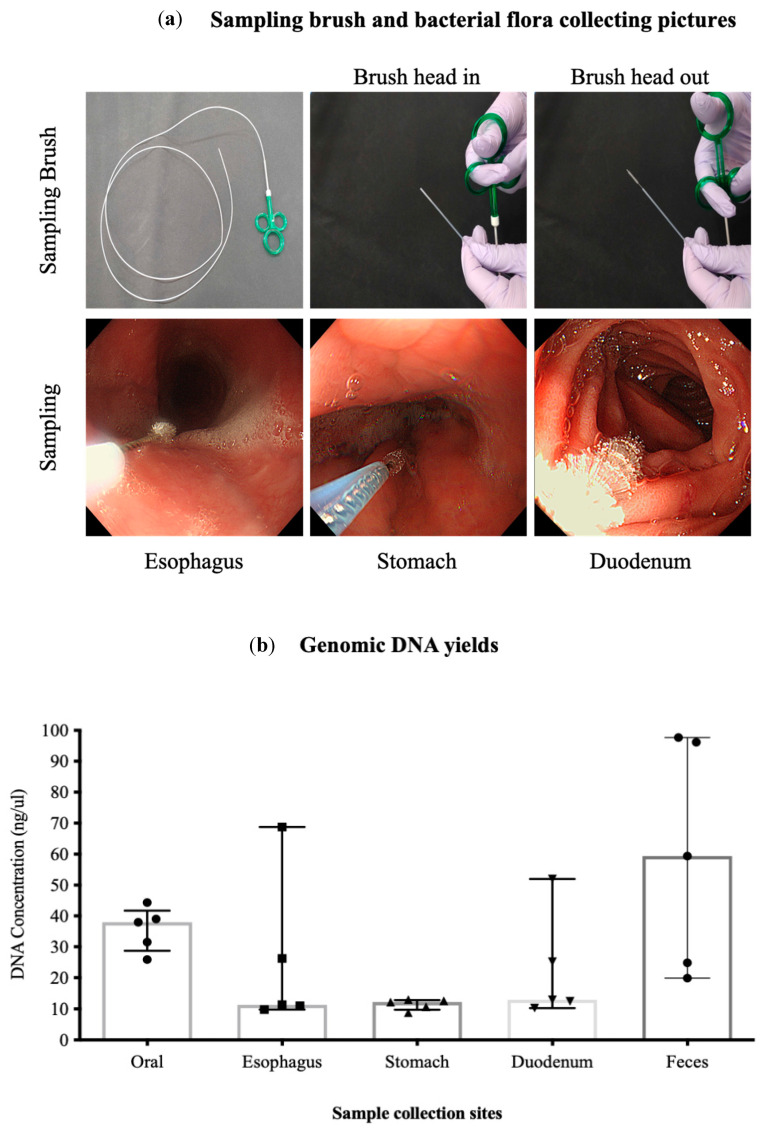
(**a**) Sampling brush and bacterial flora collection. The top row of this figure shows the sampling brush used to collect bacterial samples, manufactured by Cook Medical. This brush was used to ensure consistent and efficient sample collection across multiple sites. The bottom row shows photographs of each organ at the time of sampling. These images include the oral cavity, esophagus, stomach, and duodenum, capturing the exact conditions under which the samples were collected. (**b**) Summary of DNA concentration at each site. Genomic DNA was present in all samples. The oral, esophagus, and fecal samples had sufficient DNA concentrations, whereas the stomach and duodenum samples had relatively low concentrations, making collection challenging. (**c**) Testing bacterial DNA via 16S PCR. Bacterial DNA was successfully amplified from all collected samples using 16s PCR, with PCR products identifiable at 550 bp. Although amplification was difficult for gastric and duodenal samples (indicated by red and black arrows), oral and fecal samples showed clear bands due to sufficient bacterial DNA.

**Figure 2 microorganisms-12-01324-f002:**
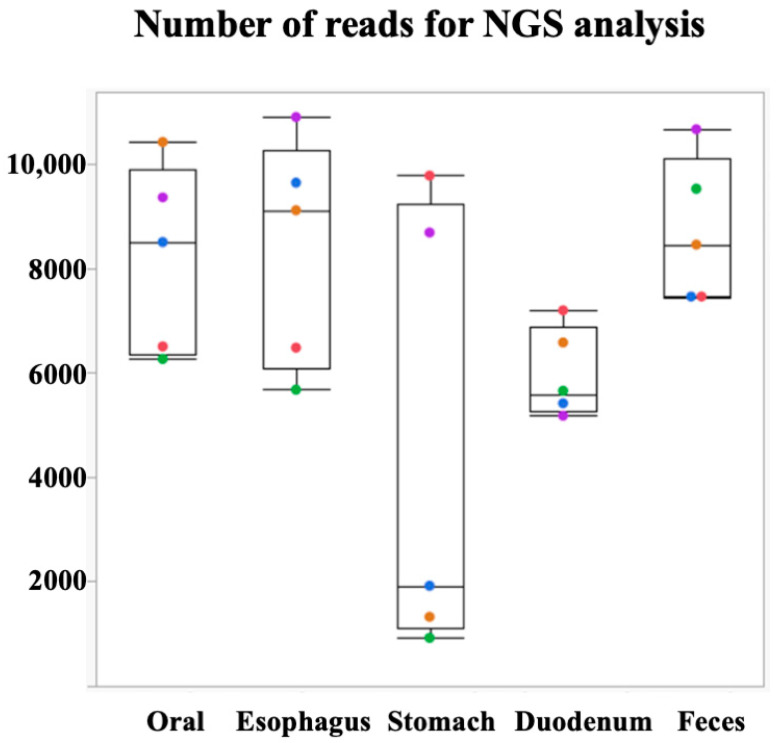
Number of reads was sufficient for NGS analysis. This graph shows that the number of reads obtained was sufficient for NGS analysis. On the y-axis, the graph represents the number of reads, which indicates the amount of bacterial DNA sequences obtained from each sample. Samples from the esophagus, stomach, and duodenum provided enough DNA to effectively assess their bacterial flora. However, in the stomach and duodenum, where conditions are characterized by high acidity or alkalinity, significant variations in read counts were observed. Some samples from these regions yielded only about a tenth of the expected reads, highlighting the challenging conditions for bacterial survival and DNA extraction in these areas.

**Figure 3 microorganisms-12-01324-f003:**
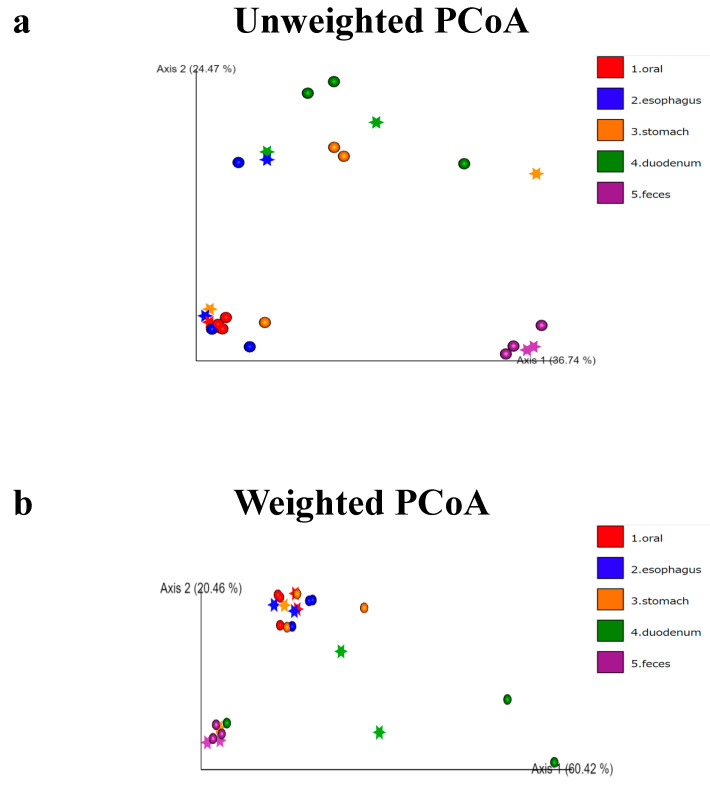
PCoA analysis. A principal component analysis (PCoA) plot is shown. In this plot, each point represents a sample, and the data are projected onto a two-dimensional space by Principal Component 1 (PC1) and Principal Component 2 (PC2). Clusters in different colors represent different groups, making it possible to visualize similarities between samples and the separation between groups. In the unweighted PCoA (**a**), the stomach samples are diverse but form clusters in the other groups. In the weighted PCoA (**b**), the duodenal samples are varied, but the oral cavity, esophagus, and stomach samples form almost identical clusters.

**Figure 4 microorganisms-12-01324-f004:**
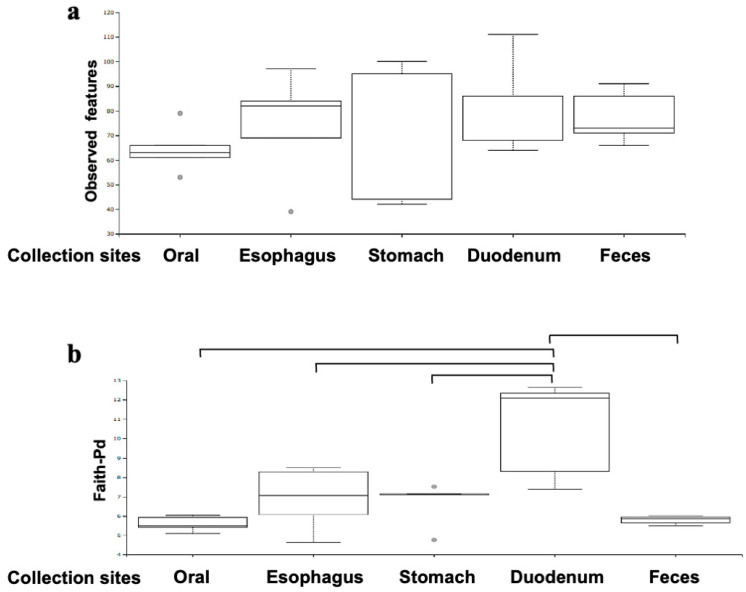
Comparative analysis of microbial richness and diversity across GI sites. The combined results of the five subjects show that the richness (the number of characteristics observed) of the oral flora is concentrated in a narrow range; meanwhile, the results for gastric mucus vary significantly between samples. Statistically, there are no significant differences among the five groups due to the small number of patients (**a**). The duodenum was found to be phylogenetically diverse ((**b**), *p* = 0.049 Willcoxon with Bonferroni correction).

**Figure 5 microorganisms-12-01324-f005:**
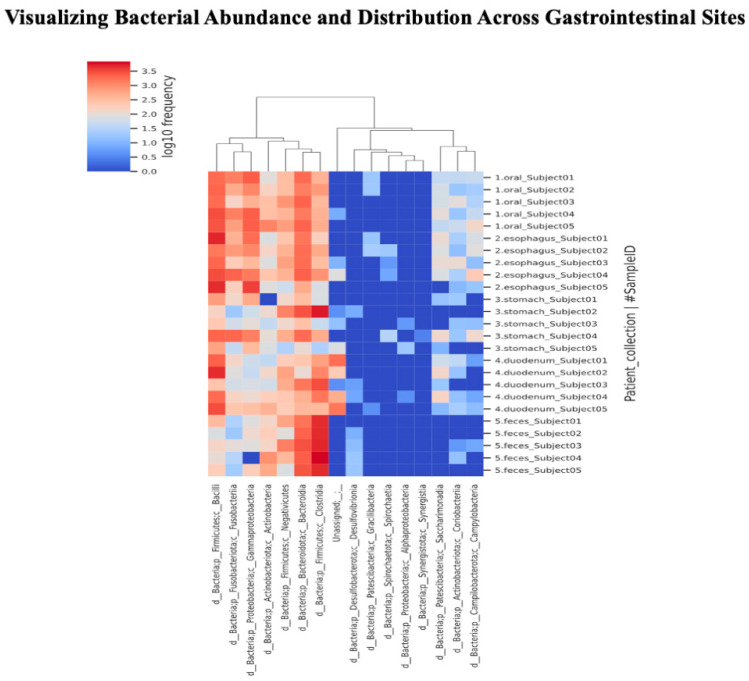
Visualizing bacterial abundance and distribution across gastrointestinal sites. The darker the red, the more abundant the species, and the darker the blue, the less abundant the species. The rows show samples sorted according to collection sites, while the columns contain assigned bacterial species. When a tree is drawn on the top of the heat map, regardless of the collection site, it can be seen that species with biological similarity are generally clustered according to site. It can also be visually confirmed that the distribution of the bacterial flora in the fecal samples (the bottom five rows) differs from the distribution of the intestinal flora in the upper gastrointestinal tract, which extends from the oral cavity to the duodenum.

**Figure 6 microorganisms-12-01324-f006:**
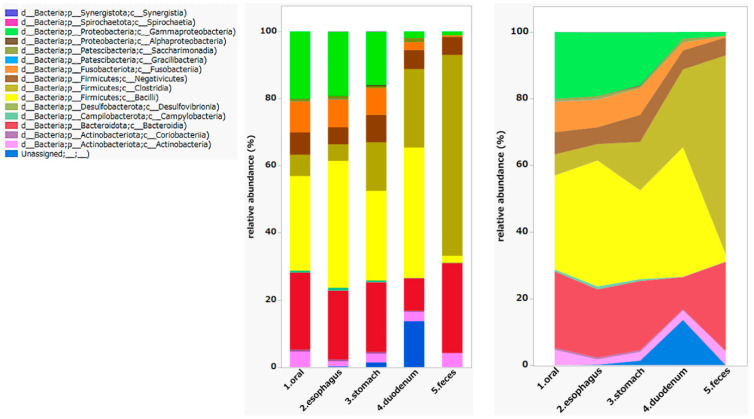
Class-level microbiome. The charts provided include a stacked bar chart and an area chart. Stacked Bar Chart: The left graph illustrates the relative abundance of different bacterial taxa in different gastrointestinal sites, such as the oral cavity, esophagus, stomach, duodenum, and feces. Each color corresponds to a distinct bacterial group, with the height of each colored section representing the proportion of that group in the sample from each site. Area Chart: The right graph similarly shows the relative abundance of bacterial taxa but uses continuous areas to show how the proportions of different bacterial groups change across sites. The distribution of microorganisms varies throughout the GI tract: *Fusobacteria* and *Bacilli* are dominant in the upper GI tract, while *Clostridia* and *Bacteroidia* are dominant in the feces. *Negativicutes* and *Actinobacteria* are present throughout the digestive tract.

**Table 1 microorganisms-12-01324-t001:** A total of 25 samples were collected from 5 healthy Japanese individuals. All collected samples contained bacterial DNA with 260/280 ratio, as shown in Table 1. However, the DNA concentration of gastric and duodenal samples was relatively low, and that of esophageal and duodenal samples varied in wide range.

Sample ID	Age	Gender	Disease Stats	Collection Site	DNA Concentration (ng/µL)	DNA 260/280 Ratio
Subject 1	42	Male	health	oral	31.5	2.45
				esophagus	26.2	2.49
				stomach	10.7	2.7
				duodenum	52	1.99
				feces	24.8	2.3
Subject 2	52	Male	health	oral	38	2.14
				esophagus	11	2.57
				stomach	12.6	2.3
				duodenum	12.4	2.29
				feces	97.7	2.01
Subject 3	54	Male	health	oral	44.4	2.11
				esophagus	11.3	2.54
				stomach	12.2	2.28
				duodenum	10.3	2.6
				feces	96.2	2.1
Subject 4	50	Male	health	oral	25.9	2.23
				esophagus	68.8	2.1
				stomach	13	2.67
				duodenum	12.9	2.58
				feces	59.4	2.01
Subject 5	41	Female	health	oral	39	2.12
				esophagus	9.8	2.43
				stomach	8.8	2.29
				duodenum	25.2	2.05
				feces	19.9	1.53

## Data Availability

The original contributions presented in the study are included in the article/Appendix A, further inquiries can be directed to the corresponding authors.

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
