# Peer review of "Sequential Sampling of the Gastrointestinal Tract to Characterize the Entire Digestive Microbiome in Japanese Subjects"

_microorganisms, 2024, doi:10.3390/microorganisms12071324_

Round 1

Reviewer 1 Report

Comments and Suggestions for Authors

In this study, Ishizawa et al. presented a comprehensive analysis of the gastrointestinal microbiome across the entire digestive tract of 5 Japanese subjects, revealing significant microbial diversity, particularly in the duodenum, and demonstrating the feasibility of sequential sampling for future research. This study proposed a novel method for sampling microbiota, but there are a few issues with the manuscript. Here are some comments on this study:

1.        Please define the abbreviation when it first appears, such as “EGD”.

2.        The figures are too low of a resolution and nearly impossible to read.

3.        Line 131 “using 130 1,000 sequences per sample” and line 258 “The number of reads obtained from the NGS was approximately 1000, which is con-258 sidered good”, I disagree with the author's stated view that 1000 sequences per sample is enough. Through Figure 2, it could be noticed that only 3 stomach samples had 1000 features and the other samples had feature numbers higher than 5000. If the number of features is rarefied to 1000, it will result in the loss of many features in most of samples. I consider a reasonable way to do this is to exclude low feature samples.

4.        Table 1 appears not to be neatly organized and the authors are advised to refine it, e.g. by combining the Sample ID and Disease stats. What is more, the table legend is missing and the results of DNA quality are needed.

5.        Figure 1 C, the ladder information is required and labeled.

6.        Figure 4 b, is there a multiple comparison such as Tukey?

7.        Line 261 the names of bacteria genus “Fusobacteria and Bacillus” are proposed to be italicized.

8.        It would be useful for the authors to update the references.

Author Response

Reviwer 1

Comments and Suggestions for Authors

In this study, Ishizawa et al. presented a comprehensive analysis of the gastrointestinal microbiome across the entire digestive tract of 5 Japanese subjects, revealing significant microbial diversity, particularly in the duodenum, and demonstrating the feasibility of sequential sampling for future research. This study proposed a novel method for sampling microbiota, but there are a few issues with the manuscript. Here are some comments on this study:

  • Please define the abbreviation when it first appears, such as “EGD”.
    • Thank you for the comment. I added amendment of EGD stand for ‘Esophagogastroduodenoscopy’ in line 81.
  • The figures are too low of a resolution and nearly impossible to read.
    • Thank you for your comment, I have updated all the figures with a higher resolution.
  • Line 131 “using 130 1,000 sequences per sample” and line 258 “The number of reads obtained from the NGS was approximately 1000, which is con-258 sidered good”, I disagree with the author's stated view that 1000 sequences per sample is enough. Through Figure 2, it could be noticed that only 3 stomach samples had 1000 features and the other samples had feature numbers higher than 5000. If the number of features is rarefied to 1000, it will result in the loss of many features in most of samples. I consider a reasonable way to do this is to exclude low feature samples.
    • Thank you for your comment. It is very important to determine the rarefaction of sequence reads per sample. We chose 1000 reads based on α-rarefaction analysis. We plotted the diversity of α-rarefaction and checked that each sample reached the plateaus around 1000 reads, based on this figure we think that 1000 is a reasonable number. Nevertheless, we also plotted the number 5000 reads as suggested, and have included these figures in the supplementary data (heatmap and stacked area plot of bacterial relative abundance).
  • Table 1 appears not to be neatly organized and the authors are advised to refine it, e.g. by combining the Sample ID and Disease stats. What is more, the table legend is missing and the results of DNA quality are needed.
    • Thank you for the comment. We have added the table legend. We also changed the layout of the table and added the DNA 260/280 ratio as a DNA quality marker. The table legend is shown below.
    • A total of 25 samples were collected from five healthy Japanese individuals. All collected samples contained bacterial DNA with 260/280 ratio as shown in Table 1. However, the DNA concentration of gastric and duodenal samples was relatively low, esophageal and duodenal samples varied with wide range.Figure 1 C, the ladder information is required and labeled.
  • Thank you for the comment. The ladder has been added to the Figure 1C. Please have a check on it.
  • Figure 4 b, is there a multiple comparison such as Tukey?
    • Thank you for the comment. Yes, we performed the analysis using Wilcoxon with Bonfferoni correction and have updated in the figure legend.
  • Line 261 the names of bacteria genus “Fusobacteria and Bacillus” are proposed to be italicized.
    • Thank you for the comment. We corrected “Fusobacteria and Bacillus” toFusobacteria and Bacillus, and all other taxonomy words.
  • It would be useful for the authors to update the references.
    • Tank for the comment. This is very important point to update the current references, we have put 5 more new papers.

[1–5]

  1. Yamaki, K.; Tamahara, T.; Washio, J.; Sato, T.; Shimizu, R.; Yamada, S. Intracanal Microbiome Profiles of Two Apical Periodontitis Cases in One Patient: A Comparison with Saliva and Plaque Profiles. Clin. Exp. Dent. Res. 2024, 10, e862, doi:10.1002/cre2.862.
  2. Saito, S.; Aoki, Y.; Tamahara, T.; Goto, M.; Matsui, H.; Kawashima, J.; Danjoh, I.; Hozawa, A.; Kuriyama, S.; Suzuki, Y.; et al. Oral Microbiome Analysis in Prospective Genome Cohort Studies of the Tohoku Medical Megabank Project. Front. Cell. Infect. Microbiol. 2021, 10, 604596, doi:10.3389/fcimb.2020.604596.
  3. Kawana, T.; Imoto, H.; Tanaka, N.; Tsuchiya, T.; Yamamura, A.; Saijo, F.; Maekawa, M.; Tamahara, T.; Shimizu, R.; Nakagawa, K.; et al. The Significance of Bile in the Biliopancreatic Limb on Metabolic Improvement After Duodenal-Jejunal Bypass. Obes. Surg. 2024, 34, 1665–1673, doi:10.1007/s11695-024-07176-7.
  4. Ikeda, H.; Ihara, E.; Takeya, K.; Mukai, K.; Onimaru, M.; Ouchida, K.; Hata, Y.; Bai, X.; Tanaka, Y.; Sasaki, T.; et al. The Interplay between Alterations in Esophageal Microbiota Associated with Th17 Immune Response and Impaired LC20 Phosphorylation in Achalasia. J. Gastroenterol. 2024, 59, 361–375, doi:10.1007/s00535-024-02088-w.
  5. Duvallet, C.; Gibbons, S.M.; Gurry, T.; Irizarry, R.A.; Alm, E.J. Meta-Analysis of Gut Microbiome Studies Identifies Disease-Specific and Shared Responses. Nat. Commun. 2017, 8, 1784, doi:10.1038/s41467-017-01973-8.
  6. NIH Human Microbiome Project Defines Normal Bacterial Makeup of the Body | National Institutes of Health (NIH) Available online: https://www.nih.gov/news-events/news-releases/nih-human-microbiome-project-defines-normal-bacterial-makeup-body(accessed on 20 June 2024).

Reviewer 2 Report

Comments and Suggestions for Authors

This study aims to demonstrates the possibility to continuously collect microbiome samples throughout the human digestive tract to reveal the complexity of GI microbiota.

It is significant in clinical. The most important weakness of this manuscript is the sample size is too small. More cases of investigation can lead to more solid conclusion.

In result section: From Figure 3-5, more details need to be added. They are kind of confusing and not clear to readers. And the figures are blur and hard to read.

Other minors:

Line 38: Is Bacterioidia a type error of bacteroidia?

Line 43: Detail the abbr. EGD for the first appear.

Author Response

Reviewer 2

Comments and Suggestions for Authors

This study aims to demonstrates the possibility to continuously collect microbiome samples throughout the human digestive tract to reveal the complexity of GI microbiota.

It is significant in clinical. The most important weakness of this manuscript is the sample size is too small. More cases of investigation can lead to more solid conclusion.

In result section: From Figure 3-5, more details need to be added. They are kind of confusing and not clear to readers. And the figures are blur and hard to read.

              Thank you for the very important comments.

  1. We agree that the sample size of this study is very small. This study represents an initial effort to comprehensively observe the entire gastrointestinal tract. We are actively continuing to enroll more patients to strengthen our findings and draw more robust conclusions as noted.
  2. Thank you for the comment, we add more explanation through Fig 3 to Fig 5. Please check the figure Legends.

Other minors:

Line 38: Is Bacterioidia a type error of bacteroidia?

              Thank you for the correction of the type error. Bacteriodia is correct.

Line 43: Detail the abbr. EGD for the first appear.

  • Thank you for the comment. I added amendment of EGD stand for ‘Esophagogastroduodenoscopy’ in line 81.

Round 2

Reviewer 1 Report

Comments and Suggestions for Authors

I thank the authors for addressing all my comments.